# Resistance to Anti-HER2 Therapies in Gastrointestinal Malignancies

**DOI:** 10.3390/cancers16162854

**Published:** 2024-08-15

**Authors:** Christiana Mo, Michelle Sterpi, Hyein Jeon, Fernand Bteich

**Affiliations:** 1Department of Medical Oncology, Albert Einstein College of Medicine, Bronx, NY 10461, USA; cmo3@montefiore.org (C.M.); msterpi@montefiore.org (M.S.); hyjeon@montefiore.org (H.J.); 2Department of Medical Oncology, Montefiore Medical Center, Bronx, NY 10467, USA

**Keywords:** HER2, gastro/gastroesophageal cancer, colorectal cancer, cancer therapeutics, resistance

## Abstract

**Simple Summary:**

For many years, trastuzumab has remained a cornerstone in the treatment of HER2-positive cancers. However, primary and acquired resistance to trastuzumab and other HER2-targeted agents commonly prevents durable treatment responses in patients. The recent breakthroughs in HER2-targeted therapies for gastrointestinal malignancies necessitate a deeper understanding of resistance mechanisms and ways to circumvent them. This review explores the different classifications of HER2 status, emerging anti-HER2 agents for various GI cancers (esophagogastric, colorectal, biliary tract, and small bowel), and recent significant clinical trials, while examining potential resistance pathways. Additionally, the review will touch upon new techniques and agents which may be used for detecting and overcoming resistance.

**Abstract:**

Human epidermal growth factor 2 (HER2) is a tyrosine kinase receptor that interacts with multiple signaling pathways related to cellular growth and proliferation. Overexpression or amplification of HER2 is linked to various malignancies, and there have been decades of research dedicated to targeting HER2. Despite the landmark ToGA trial, progress in HER2-positive gastrointestinal malignancies has been hampered by drug resistance. This review examines current HER2 expression patterns and therapies for gastroesophageal, colorectal, biliary tract, and small bowel cancers, while dissecting potential resistance mechanisms that limit treatment effectiveness.

## 1. Introduction

Human epidermal growth factor receptor 2 (HER2) is a protein encoded by the proto-oncogene *ERBB2* on chromosome 17. It is a 185 kDa transmembrane tyrosine kinase receptor that belongs to the epidermal growth factor receptor family, which consists of HER1 (EGFR), HER2, HER3, and HER4. HER2 is expressed in various normal tissues, including breast, gastrointestinal tract, kidney, and heart, and is primarily involved in cell growth [1]. However, the aberrant expression of HER2 in the gastrointestinal tract can participate in the pathogenesis of multiple malignancies, with gastric/gastroesophageal adenocarcinomas being the most common. HER2 amplification or overexpression can also be found, albeit to a lesser degree, in colorectal cancer, biliary tract cancers, and small bowel adenocarcinomas [2,3,4,5].

Similar to other receptor tyrosine kinases (RTKs), HER2 comprises an extracellular domain, a transmembrane lipophilic segment, and an intracellular tyrosine kinase domain. However, HER2 does not have a specific ligand and instead mediates signal transduction through preferential heterodimerization with other HER family proteins, such as HER3, therefore forming the most effective signaling dimer out of all possible pairings. [6,7]. This dimerization leads to autophosphorylation of its tyrosine residues in the cytoplastic domain and the activation of two major intracellular signaling cascades—the mitogen-activated protein kinase (MAPK)/Ras and the phosphatidylinositol 3′-kinase (PI3K)/AKT pathways [8]. MAPK/Ras/Raf/ERK predominately regulates cell mitosis while the PI3K/AKT pathway influences cell proliferation and apoptosis. Heterodimers with HER2 have a stronger signaling potential than standalone homodimers or when complexed with other family members [7]. Therefore, HER2 overexpression and amplification result in the dysregulation of apoptosis, excessive cell growth, angiogenesis, and eventual tumorigenesis.

Defining HER2 positivity has been challenging as the degree and the pattern of HER2 expression vary widely between different tumor types, leading to multiple definitions for “HER2-positive”. The diagnostic criteria for HER2 expression in GI malignancies were initially derived from multiple large phase II and phase III trials in breast cancer. However, key differences in HER2 expression were found between breast and GI malignancies. In gastric cancer, there is increased heterogeneity, variations in HER2 expression, based on anatomic location and an incomplete basolateral membrane staining [9,10]. A large-scale validation study established the current guidelines for HER2 testing in gastric cancer, which was then utilized for colorectal, biliary tract, and small bowel cancer [11]. Currently, HER2 expression is primarily assessed by immunohistochemistry (IHC) or in situ hybridization (ISH) assays on biopsy or surgical specimens. HER2 protein expression is classified as negative (0–1+), equivocal (2+), or positive (3+) by IHC based on the extent and pattern of membrane staining. ISH evaluates the number of HER2 gene copies and dichotomous results are reported as positive or negative. The initial assessment is typically performed with IHC, and then ISH is recommended for tumor samples with IHC 2+. HER2-positivity is defined as IHC 3+ or IHC 2+/ISH positive, as these were the patients found to benefit the most from HER2-directed therapy [12]. An emerging reclassification of HER2 protein expression through IHC adopts the HER2-zero (IHC 0) and HER2-low (IHC 1+ or 2+ with negative ISH) labels. This categorization originated in breast cancer following the clinical benefit of trastuzumab deruxtecan (T-DXd) for HER2-low patients. It is now rapidly expanding to include colorectal and gastric cancers with the advent of new therapies. Although definitive benefits have not yet been established for colorectal cancer, a phase II study has reported promising preliminary results for HER2-low gastroesophageal and gastric cancers [13,14].

It is worth nothing that there are limitations associated with IHC classification due to intratumoral heterogeneity, incomplete staining, and variability from subjective interpretation [12]. Latest advances have elicited alternative tools to determine HER2 amplification, such as next-generation sequencing, comprehensive genome sequencing, and liquid biopsies. The concordance between IHC/FISH and the newer sequencing techniques is higher in colon cancer than gastric, with some studies showing rates as high as 80% in colon cancer [15,16,17]. For colon cancer, liquid biopsies were able to detect ERBB2 alterations in circulating tumor DNA (ctDNA) that were confirmed in tissue samples, suggesting that HER2 has potential as a plasma biomarker. Additionally, response to HER2-directed therapy coincided with decreased ctDNA allele fractions [16,18,19].

The recent years have brought great strides in testing and treatment to improve clinical outcomes for HER2-positive gastrointestinal malignancies. However, since the approval of trastuzumab, the oldest drug targeting HER2, it was rapidly evident that resistance is inevitable, and patients eventually have disease progression. Trastuzumab resistance is well documented in breast cancer and has led to swift progress in the development of other HER2-directed agents. However, the same progress has been substantially slower in GI malignancies. In this review, we will discuss new anti-HER2 therapies or regimens in gastrointestinal malignancies, their known mechanisms of resistance as well as future directions to help address the remaining unmet needs.

## 2. HER2 Targeting Modalities

The landscape of HER2-targeted treatments has evolved significantly since the discovery of the ERBB2 gene in 1984. Specifically in the realm of gastrointestinal malignancies, the success of the ToGA trial has spearheaded the advancements in HER2-positive gastric/gastroesophageal junction (G/GEJ) adenocarcinomas [20]. Ongoing research aims to enhance outcomes in gastrointestinal cancers through HER2-targeted agents like monoclonal antibodies, small molecular kinase inhibitors, bispecific antibodies, antibody–drug conjugates, cellular therapies, and other immune-modulating strategies.

### 2.1. Monoclonal Antibodies

Various intravenous HER2-targeted monoclonal antibodies, such as trastuzumab, pertuzumab, and margetuximab, have been subject to clinical trials in patients with HER2-positive gastrointestinal adenocarcinomas. These studies resulted in the approval of trastuzumab in HER2-expressing G/GEJ cancers and the use of trastuzumab with pertuzumab in HER2-amplified/RAS-BRAF wild-type colorectal cancer (CRC) [21,22,23].

Trastuzumab, a humanized mAb designed to target HER2, binds to domain IV of HER2 near the juxtamembrane site. While the extracellular mechanisms of action of trastuzumab are well understood, including its immune effector functions such as CDC and CDCC, its intracellular actions are still being investigated [24,25,26]. These intracellular mechanisms may include inhibiting HER2-mediated cell signaling pathways like AKT and ERK, suppressing HER2 cleavage, interfering with DNA damage repair processes and inhibiting angiogenesis by reducing VEGF production [27,28,29,30,31,32,33,34,35,36,37].

Pertuzumab, a fully humanized recombinant mAb, inhibits HER2 dimerization by specifically interacting with subdomain II of the HER2 extracellular domain. This interaction blocks a binding pocket crucial for receptor dimerization, thereby inhibiting HER2 dimerization and downstream signaling [38].

While not yet approved for clinical use in gastrointestinal malignancies, margetuximab represents an Fc-optimized mAb targeting HER2 that has demonstrated promising outcomes in HER2-positive solid tumors, notably in gastric cancer [39]. Margetuximab was engineered to enhance binding to the activating Fcγ receptor CD16A while reducing binding to the inhibitory FcγR CD32B, improving immune responses against HER2-overexpressing cells and potentially synergizing with checkpoint inhibitors [39,40,41,42,43].

### 2.2. Tyrosine Kinase Inhibitors

Tyrosine kinase inhibitors (TKIs) are orally administered small molecules that target the intracellular catalytic kinase domain of HER2 by competing with ATP, inhibiting phosphorylation and blocking downstream signaling cascades [44]. While TKIs have shown efficacy in metastatic colorectal cancer (CRC), they have not yet been successful in HER2-positive gastric cancer (GC).

Lapatinib, an oral TKI that inhibits EGFR and HER2, exhibits activity in HER2-driven tumors resistant to trastuzumab, by blocking HER2 and insulin-like growth factor 1 receptor (IGF-1R) crosstalk [45].

Tucatinib, a potent HER2-specific TKI, inhibits HER2 and HER3 phosphorylation, suppressing MAPK and AKT signaling [46]. It has shown anti-tumor efficacy in CRC models, including HER2 mutations conferring trastuzumab resistance, suggesting a synergistic effect when combined with trastuzumab in chemo-refractory patients [23,46]. The promising results from these studies have led to the approval of this regimen in this setting. The combination of tucatinib and trastuzumab is also being explored in biliary tract cancers (BTC), where it has shown a cORR of 46.7% in pretreated HER2+ metastatic patients in a phase II trial [47].

Furthermore, pyrotinib is an orally administered, irreversible pan-HER TKI targeting EGFR, HER2, and HER4 [48,49]. It is under investigation, in combination with chemotherapy, in the second-line setting of metastatic GC [50].

### 2.3. Antibody–Drug Conjugates

Antibody–drug conjugates (ADCs) serve as a targeted therapy approach by combining mAb with cytotoxic drugs to deliver potent anti-cancer agents specifically to malignant cells, thereby reducing systemic toxicity. They contain a tumor-targeting antibody covalently bound to a cytotoxic drug (payload) via a synthetic linker [51]. In the context of gastrointestinal cancers, particular ADCs, such as trastuzumab deruxtecan (T-DXd) and RC48-ADC, have shown promising results.

T-DXd has thus far impacted the treatment landscape of gastrointestinal cancers, notably GC [52]. This ADC utilizes a HER2-targeting antibody (trastuzumab) as a vehicle to deliver a topoisomerase I inhibitor (deruxtecan) directly to HER2-expressing cancer cells [53]. Clinical trials have demonstrated its efficacy in advanced HER2-positive tumors, leading to its FDA approval for HER2-positive GC/GEJ adenocarcinoma after prior exposure to trastuzumab with chemotherapy in the first-line setting.

Most recently, the DESTINY-PanTumor02 and DESTINY-CRC02 phase II trials led to the first tumor agnostic approval of a HER2-directed therapy by the FDA in HER2-positive IHC 3+ tumors [54,55].

Similarly, disitamab vedotin (formerly known as RC48) presents a tailored approach to HER2-positive gastrointestinal cancers. This ADC consists of an anti-HER2 mAb linked to monomethyl auristatin E (MMAE) and has shown a favorable safety profile and promising anti-tumor activity in late-stage HER2-positive solid tumors, including GC. The overall response rate was 24.8%, with a median progression-free survival of 4.1 months and a median overall survival of 7.9 months [56].

### 2.4. Bispecific Antibodies

Bispecific antibodies (bsAbs) represent a diverse class of molecules designed to target two different epitopes or antigens simultaneously, offering unique advantages in therapeutic applications. These molecules can range from simplistic structures with two linked antigen-binding fragments to more complex immunoglobulin G (IgG)-like formats, providing flexibility in targeting different cellular pathways [57].

Zanidatamab (ZW25) is a bispecific antibody that binds to non-overlapping epitopes of HER2, specifically targeting extracellular domains II and IV. Through its biparatopic binding mechanism, ZW25 induces HER2-receptor clustering, internalization, and downregulation, effectively inhibiting both growth factor-dependent and -independent tumor cell proliferation [58]. Furthermore, this mAb triggers ADCC and CDC, engaging multiple pathways to attack cancer cells. These distinctive properties position ZW25 as a promising option in the treatment of gastrointestinal cancers, particularly in HER2-positive GEJ adenocarcinomas and HER2-positive biliary tract cancers (BTC). Supported by its ongoing clinical trials and promising preliminary results, ZW25 has garnered fast track designation by the FDA for first-line therapy in combination with standard of care chemotherapy in GEJ adenocarcinoma and as single agent for refractory BTC [59,60,61].

## 3. HER2 in GI Malignancies: Specific Insights into Different Gastrointestinal Cancers

### 3.1. Gastric and GEJ Adenocarcinoma

HER2 has been widely studied in G/GEJ adenocarcinomas since its discovery, and it has key clinical implications for their management. HER2 overexpression or amplification occurs in approximately 7.3–20.2% of patients with G/GEJ adenocarcinoma, with varying rates in different countries. HER2 positivity is more frequently associated with the intestinal subtype and tumors arising from the gastroesophageal junction [62,63]. The prognostic role of HER2 remains controversial, but a larger number of studies indicate that HER2 amplification is associated with more aggressive tumor behavior and a higher recurrence rate [2,64].

The current standard of care for first-line treatment of HER2-positive G/GEJ cancer is a combination of chemotherapy and trastuzumab, which is based on the pivotal phase III ToGA trial. The addition of trastuzumab resulted in a significant survival benefit (median OS 13.8 months vs. 11.1 months; *p* = 0.046) (Table 1) [20]. After the success of the ToGA trial, later attempts to target HER2 with other agents proved to be disappointing. While pertuzumab, lapatinib, and T-DM1 are approved in HER2-positive breast cancer, they did not demonstrate the same efficacy in HER2-positive G/GEJ cancer. In the first-line setting, the JACOB trial evaluated the addition of pertuzumab to the standard ToGA regimen, but there was no significant improvement in OS [65]. For second-line treatment, the T-ACT (trastuzumab with paclitaxel), GATSBY (T-DM1), and TyTan (lapatinib) trials all failed to demonstrate a significant survival benefit when compared to paclitaxel [66,67,68]. The lack of positive data from these large-scale trials has been puzzling and discouraging.

Fortunately, more recent advances in targeted and immune therapies have led to the development of new agents and combinations for targeting HER2-positive G/GEJ cancer. The results from the phase III KEYNOTE-811 study demonstrated the efficacy of adding pembrolizumab to the standard regimen of trastuzumab and chemotherapy (mPFS 10.0 vs. 8.1 months, mOS 20.0 vs. 16.8 months) in allcomers. Interestingly, subgroup analysis revealed only patients with a combined positive score (CPS) ≥ 1, indicating PD-L1 positivity, had statistically improved mPFS (10.9 vs. 7.3 months) and mOS (20.0 vs. 15.7 months) [69]. The phase II DESTINY-Gastric01 showed significant improvement with T-DXd compared to standard chemotherapy in terms of ORR (51.3 vs. 14.3%) and mOS (12.5 vs. 8.4 months) for patients with heavily pretreated HER2-positive G/GEJ cancers, leading to its approval as second-line therapy [70]. Other ongoing efforts include the phase III MAHOGANY trial studying first-line margetuximab with either retifanlimab (a PD-1 inhibitor) or tebotelimab (a PD-1xLAG-3 bispecific) for metastatic HER2-positive G/GEJ cancers, as well as the phase II/III MOUNTAINEER-2 study evaluating the second-line chemotherapy-free association of tucatinib and trastuzumab [43,71].

**Table 1 cancers-16-02854-t001:** FDA approved HER2directed treatments for HER2 positive Gastrointestinal Cancers.

Cancer Type	Stage	Treatment	Targeting Modalities	Indications	Trial
Gastric/GEJ	Unresectable, locally advanced or metastatic	Trastuzumab + Fluoropyrimidine (5-FU or capecitabine) + Cisplatin	Monoclonal antibody	1st line	ToGA [20]
	Unresectable, locally advanced or metastatic	Fam-trastuzumab deruxtecan-nxki	ADC	2nd line	DESTINY-Gastric01 [70]
Colorectal Cancer	Chemotherapy refractory RAS wild-type unresectable or metastatic	Tucatinib + trastuzumab	TKI (tucatinib), monoclonal antibody (trastuzumab)	2nd line	MOUNTAINEER [72]
	Unresectable or metastatic	Fam-trastuzumab deruxtecan-nxki	ADC	3rd line	DESTINY-CRC01 [73] DESTINY-CRC02 [74]

ADC, antibody drug conjugates; GEJ, gastro-esophageal junction cancer; TKI, tyrosine kinase inhibitors.

### 3.2. Colorectal Cancer

In the landscape of CRCs, approximately 3–5% of cases exhibit HER2 amplification or protein overexpression [3]. Notably, HER2 amplification progressively increases in prevalence from the right- to left-sided CRC tumors [75] and is predominantly associated with tumors that are wild-type for KRAS and BRAF genes. This molecular profile has significant clinical implications, including a higher incidence of central nervous system (CNS) metastases and diminished efficacy of anti-EGFR therapies, leading to poorer clinical outcomes [76].

Recent advances in targeted therapies have been underscored by the MOUNTAINEER trial, a seminal phase II study that supported the FDA accelerated approval of tucatinib and trastuzumab as a second-line treatment for chemotherapy-refractory, HER2-positive, RAS wild-type unresectable, or metastatic CRC [72]. This regimen achieved a median overall survival of 24.1 months, an objective response rate (ORR) of 38.1%, and a median duration of response of 12.4 months [23]. The MOUNTAINEER-3 trial is currently underway to evaluate using first-line trastuzumab, tucatinib, and the FOLFOX regimen versus standard of care chemotherapy. This endeavor, if successful, will also be an important addition to further improve upon the standard of care with HER2-directed therapies in earlier lines of treatment [77].

MyPathway is a multi-basket trial using pertuzumab and trastuzumab, a combination of antibodies commonly used for breast cancer, for tissue-agnostic cohort of patients with HER2-altered solid tumors. This trial demonstrated a median OS of 15.5 months, ORR of 31.9%, and a median PFS of 4.1 months in the HER2-amplified or overexpressed colorectal cancer group, while a particularly higher ORR of 41% was seen amongst patients across all cancer types with IHC staining of 3+ [78]. Similarly, other trials using trastuzumab and pertuzumab, such as the phase II TRIUMPH trial, showed a similar ORR of 30% in tissue positive for HER2 amplification versus 28% in patients with ctDNA-positive HER2 amplification [18]. The HERACLES trial is a phase II study investigating trastuzumab and lapatinib, a TKI targeting the HER2 receptor, to potentially overcome resistance in a combinatory effort in HER2-positive KRAS exon 2 wild-type metastatic CRC. The trial included patients with HER2 amplification or 3+ IHC staining who were refractory to multiple standard treatments. The combination yielded an ORR of 30%, with about 21 weeks of median progression-free survival [79].

Additionally, the phase II studies DESTINY-CRC01 and DESTINY-CRC02 are evaluating the efficacy of trastuzumab deruxtecan (T-DXd), a HER2 ADC. In DESTINY-CRC01, a cohort with positive HER2 alterations showed a promising ORR of 45.3%. Median PFS and OS were 6.9 and 15.5 months, respectively [73]. DESTINY-CRC02 is currently ongoing. It utilizes two different dosing regimens with a promising preliminary result of ORR nearing 37.8% in the 5.4 mg/kg arm, with the majority of patients being HER2 IHC 3+ and RAS wild-type tumors [74].

### 3.3. Biliary Tract Cancers

HER2-targeted therapies hold promise for biliary tract cancers, where HER2 expression and amplification rates are notably high at 23–32% and 17–23%, respectively [4], with a mutation rate of 6.2% according to meta-analyses [80]. The highest amplification rates are seen in extrahepatic cholangiocarcinoma and gallbladder cancer at 20% and 17%, respectively, with the highest mutation rate in extrahepatic cholangiocarcinoma at 5.2% [81]. Currently, no HER2-directed therapies are approved for biliary cancers, but the aggressive nature of these tumors has spurred exploration into targeting genomic alterations, including HER2.

Several studies are investigating various combinations of trastuzumab for biliary tract cancers. The MyPathway trial, a multi-basket study, included HER2-overexpressing or amplified biliary tract cancers and reported a median OS of 12.9 months, an ORR of 30.6%, with a median PFS of 4.7 months [78]. Other combination studies, such as NCT04579380 using trastuzumab and tucatinib and NCT04722133 using trastuzumab and FOLFOX chemotherapy, demonstrated ORRs of 46.7% and 29.4%, with a median OS of 12 months and 10.7 months, respectively. Conversely, the phase 2 TAB trial using trastuzumab with gemcitabine and cisplatin reported a median PFS of 7 months with a disease control rate of 80%, albeit with higher toxicity [82]. These findings underscore the potential of HER2-targeted therapies in biliary tract cancers and highlight the need for continued research and development in this area.

### 3.4. Small Bowel Adenocarcinoma

Small bowel adenocarcinoma (SBA) is a rare cancer, comprising less than 5% of all gastrointestinal malignancies [83]. The most prevalent subtype of SBA is duodenal adenocarcinoma (DA), which constitutes over 50% of SBA cases and accounts for 30% of small intestinal cancers [84]. The potential for HER2 as a therapeutic target in SBA has been suggested by multiple genetic screening and profiling studies. The incidence of HER2 genetic alterations, including mutations and amplifications, varies among different studies, ranging from 1.7% to 12% [5,85,86].

Research on SBA management, including adjuvant chemotherapy and radiation therapy, is hindered by limited study sizes and retrospective series. Due to the low prevalence of DA, clinical trials to establish optimal treatments are challenging. Consequently, patients with these malignancies are often treated with drug regimens traditionally used for colorectal or gastric cancers. This practice extends to cases of HER2-overexpressing SBA. There are isolated reports in the literature describing the use of trastuzumab in combination with standard chemotherapy in the adjuvant and neoadjuvant settings, as well as with PD-1 inhibitors in advanced disease stages [83,87]. Overall, while HER2 presents a promising target, comprehensive studies and larger prospective studies are necessary to develop and validate effective HER2-focused therapies for SBA.

## 4. Resistance to HER2-Targeted Therapies

Despite the success of the addition of trastuzumab to chemotherapy, subsequent attempts to emulate the same results from the landmark ToGA trial have fallen short. Trastuzumab beyond progression and other HER2-based therapies in the first- or second-line have failed to demonstrate a survival benefit in G/GEJ adenocarcinomas [66]. Even less is known about other HER2-expressing GI malignancies. Additionally, while many patients may initially respond to HER2 inhibition, they lack a durable response due to primary or acquired resistance. The mechanisms of resistance to HER2 therapy in GI cancers are not fully elucidated, although several potential mechanisms are under active investigation (Figure 1).

### 4.1. HER2 Heterogeneity

Intratumoral heterogeneity is frequently seen in gastric cancer, with the incidence varying across studies from 30.0 to 75.4% [62,88,89]. This wide range is likely due to multiple types of tissue sampling and the lack of standardized guidelines to assess HER2 heterogeneity. While the reasons for HER2 heterogeneity are not clearly elucidated, hypotheses include the development of HER2-amplified neoplastic clones in a largely HER2-negative tumor or the partial silencing of HER2 expression in a homogeneous HER2-positive tumor [90]. HER2 heterogeneity has also been found between primary and metastatic sites. The GASTHER1 study reported that 5.7% of patients with HER2-negative advanced gastric cancer later developed HER2-positive metastatic lesions [91].

Heterogeneity creates problems for optimal biopsy strategies and the characterization of biomarkers for targeted therapies. Additionally, guidelines differ on their recommendations for when to perform HER2 testing. The European Society of Medical Oncology (ESMO) recommends HER2 testing for metastatic GC, while the American Society of Clinical Oncology (ASCO) recommends HER2 testing on either primary or metastatic lesions for all patients with a diagnosed GC who are able to tolerate combination therapy [92,93]. In clinical practice, endoscopic biopsy specimens are the most commonly used method to determine HER2 status in GI malignancies. However, evaluation based on the minimum five to eight tissue samples raises the concern of false negatives in tumors with HER2 heterogeneity [93,94,95]. In GC, studies have reported concordance rates ranging from 74.1% to 96.1% between the biopsy and surgical resection specimens [96,97,98]. Interestingly, intratumoral heterogeneity may be less common in CRC as the majority of HER2-amplified tumors demonstrated homogenous or mosaic expression pattern. Heterogeneity was noted in only 12% of HER2-amplifed and 2% of HER2-low tumors [99].

In G/GEJ adenocarcinoma, HER2 heterogeneity is considered a poor prognostic indicator. Patients with heterogeneous HER2-positive disease had significantly shorter progression-free survival (PFS) and overall survival (OS) than those with homogenous HER2 amplification [100]. Similar findings were reported in HER2-rescued patients when compared to initially HER2-positive patients. HER2-rescued patients are those who were HER2-negative at time of diagnosis but became HER2-positive on repeat biopsy prior to starting first-line treatment [101]. There are data to support that HER2 heterogeneity is linked with co-amplification of other receptor tyrosine kinases (RTK), such as EGFR, MET, and FGFR2, promoting resistance to HER2-directed therapy [63,102].

### 4.2. Loss of HER2 Expression

Given the dynamic nature of cancer genome and intratumoral heterogeneity, loss of HER2 positivity is one of the primary causes of acquired resistance. Multiple studies have found that HER2-amplified tumors become negative on repeat biopsies after receiving HER2-based therapy. The T-ACT trial noted that 69% of GC cases had HER2 loss between first- and second-line treatment, while others have reported rates of between 14% to 60%. The disappearance of HER2 amplification is most commonly seen in IHC2+/FISH+ GC HER2-positive tumors [66,103,104,105,106].

Interestingly, the GASTHER3 study found an increase in HER2 heterogeneity from 2.9% to 21.9% in post-progression GC biopsies. A large portion of the patients who became HER2-negative after HER2-directed therapy also demonstrated new heterogeneity. This concurrent loss of HER2-amplification and gain of heterogeneity were associated with poor response to HER2 therapy and shortened PFS [105]. It is unclear what drives the overall change in HER2 expression, whether it is only the HER2-negative cells that survive or if the HER2-positive cells convert to HER2-negative with treatment. While it is currently not considered standard of care, patients who progress on first-line HER2-based therapy should be considered for repeat biopsies to determine their current HER2 status.

### 4.3. Changes in Intracellular Signaling

Another potential source of acquired resistance are changes in intracellular signaling that hamper the inhibitory effect of HER2-targeted therapy. Genomic alterations such as point mutations, amplifications, and deletions have been implicated in the activation of compensatory downstream pathways. Multiple studies have identified common amplifications in *CCNE1*, *CDK6*, *EGFR*, *MET*, and *MYC*, and mutations in the *PTEN* and *PIK3CA* gene [63,107,108].

#### 4.3.1. MET Amplification

Mesenchymal–epithelial transition factor protein (MET) is a member of the RTK family that is frequently overexpressed or co-amplified in HER2-positive GC. Rates of HER2 and MET co-expression range from 12% to 45% across varying pre-clinical studies and large scale next-generation sequencing studies [109]. Prior research has also shown that co-amplification of both MET and HER2 led to significantly poorer overall survival when compared to overexpression of either alone in GC [110,111].

Elevated levels of MET and its ligand, hepatocyte growth factor (HGF), have been implicated in the development of resistance to HER2-targeted therapies through the restimulation of the MAPK and AKT signaling pathways [112,113]. The reactivated MAPK pathway then mediates escape from HER2-blockade induced growth inhibition by allowing G1 arrested cells to enter normal cell cycle progression. Multiple pre-clinical studies involving lapatinib-resistant GC cell lines demonstrated successful reversal of the resistance with MET inhibition [112,113,114]. Similarly, studies with afatinib-resistant GC cell lines were able to overcome the resistance through MET inhibition [115,116].

Unlike with other EGFR/RTK receptors, there is no evidence of HER2–MET heterodimerization, therefore the AKT activation is likely a result of HGF binding to MET. This is supported by in vitro evidence, as the addition of HGF to MET-amplified, lapatinib-treated GC cells restored MAPK/AKT signaling [112,113]. Accordingly, high levels of HGF are associated with poor response to HER2-directed treatment and shorter survival [112].

#### 4.3.2. *PTEN* Mutation

*PTEN* (phosphate and tensin homolog) is a tumor suppressor gene encoding for an enzyme with dual-specificity phosphatase activity. It catalyzes the degradation of phosphatidylinositol-(3,4,5)-triphosphate (PIP3), which leads to the inhibition of the downstream AKT pathway. Additionally, *PTEN* plays a regulatory role in the ERK/MAPK pathway through the dephosphorylation of the focal adhesion kinase (FAK) [117]. *PTEN* interferes with cell survival, proliferation, and migration by modulating both the AKT and MAPK pathways. Trastuzumab binding stabilizes and activates *PTEN*, thereby downregulating the PI3K/AKT signaling pathway. If *PTEN* function is lost, PI3K remains constitutively active regardless of HER2 inhibition [107].

Gene mutation is the most common reason for loss of *PTEN* activity in GC, which is associated with shortened PFS and OS, as well as poor response to trastuzumab-based therapy [118,119,120,121]. One study identified *PTEN* loss in 34.5% of HER2-positive GC patients, while another found *PTEN* loss in 47.9% [118,122]. The effect of trastuzumab on cell growth inhibition and apoptosis was significantly decreased in GC cells lacking *PTEN* as AKT remained phosphorylated. Interestingly, while HER2 expression is notorious for its heterogeneity, several studies have indicated that *PTEN* deletion is homogenous in GI malignancy tissues, suggesting that it has the potential as a biomarker to evaluate the efficacy of HER2-directed therapy [122,123].

#### 4.3.3. *PIK3CA* Mutation

While less frequent than *PTEN* mutations, *PIK3CA* mutations have been implicated in the development of HER2 therapy resistance as the activating mutations lead to constituent PI3K pathway activation [107]. In a study of 264 resected GC and GEA specimens, the rate of *PIK3CA* mutations was only 5.6% [122]. However, liquid biopsies and circulating tumor cell-DNA (ctDNA) analyses in GC patients revealed higher prevalence rates, reaching up to 66.6%. It should be noted that these findings were based on much smaller sample sizes [124,125]. *PI3KCA* mutations are also implicated in resistance to HER2 blockade in CRC. ctDNA analysis of pre- and post-progression samples from the HERACLES trial found increased *PIK3CA* mutations in patients with clinical progression [126].

In vitro studies of GC cell lines with *PIK3CA* mutations demonstrated less sensitivity to lapatinib and trastuzumab. While lapatinib was able to fully inhibit HER2 phosphorylation, it had minimal effect on AKT, which is a downstream effector of PI3K [107].

#### 4.3.4. Overexpression of Tyrosine Kinases and Their Ligands

As HER2 does not have a specific binding ligand, it can form heterodimers with other RTK partners to activate the same downstream signaling pathways as the HER2 homodimer. Overexpression of EGFR and HER3 increases HER2–EGFR and HER2–HER3 heterodimer formation, allowing tumor cells to reactivate HER2 signaling with only a few HER2 molecules present [127]. Pre-clinical studies with trastuzumab-resistant GC cells had notably higher EGFR expression and dimerization with HER2 as compared to non-resistant cells [107]. Additionally, multiple HER2-family ligands such as epidermal growth factor (EGF), amphiregulin (AREG), and neuregulin 1 (NRG1) are known to be overexpressed as well in trastuzumab-resistant GC cells [128]. This increase in both HER ligands and RTKs protects cells by allowing for compensatory activation of PI3K/AKT and MAPK signaling pathway even in the presence of anti-HER2 agents.

HER3 and HER4 can form heterodimeric complexes with HER2 through the NRG1 ligand [129]. NRG1 is expressed in multiple types of cancer cells and is implicated in therapy resistance to EGFR-targeting antibody cetuximab in CRC, RAF inhibitors in BRAF mutant melanoma, and MEK inhibitors in metastatic uveal melanoma [130]. NRG1-mediated resistance to HER2-directed therapy relies on interactions between NRG1, HER3, and HER2 activity to activate PI3K/AKT/mTOR signaling pathways [129]. Additionally, HER4 has been shown to be upregulated and phosphorylated in resistant cells when compared to the sensitive parental cells in in vitro studies. Preclinical studies indicate that, despite treatment with lapatinib, there is HER3 and residual HER2 activity in the GC cell lines, which allows NRG1 to counteract lapatinib-induced cell cycle arrest and apoptosis [129].

IGF-1R and FGFR3 are other commonly overexpressed transmembrane proteins with RTK activity in GI malignancies. Phosphorylated IGF-1R proteins were found to be upregulated in trastuzumab-resistant GC cells, leading to stimulation of the PI3K/AKT and MAPK pathways despite the HER2 blockade [121]. In vitro studies demonstrated that sensitivity to trastuzumab was restored to the resistant GC cells upon treatment with an IGFR1R inhibitor [121].

Transcriptome analysis of trastuzumab-resistant GC cells found consistent amplification of both the FGFR3 gene and a gene coding for its ligand, FGF9. The increased levels of FGFR3 allowed for activation of PI3K/AKT, sustaining tumor growth and the EMT phenotype despite anti-HER2 therapy [131].

#### 4.3.5. Amplification of Cell-Cycle Mediators

Studies have shown that co-amplifications of cell-cycle mediators, such as *CDK6*, *CCND1*, and *CCNE1*, decrease the effectiveness of HER2-targeted therapy in GI malignancies [107]. In a transcriptome analysis of 83 cases of HER2-positive advanced GC treated with trastuzumab, amplifications were frequently seen in genes associated with the G1/S cell cycle checkpoint, such as CCNE1 (28.6%) and CDK6 (9.52%) [108].

*CCND1* encodes for cyclin D1, which interacts with CDK4 and CDK6 to promote cell-cycle progression. *CCNE1* encodes for cyclin E1, which forms a complex with CDK2 to promote cell-cycle progression by initiating DNA replication at the G1/S checkpoint level. It is frequently amplified in HER2-positive GC, with rates varying between 11 and 40%, and associated with worse prognosis and poor treatment response to both trastuzumab and lapatinib [108,132,133]. Similarly, higher copy number variation (CNV) for *CCNE1* was associated with shorter survival in patients treated with trastuzumab [134].

#### 4.3.6. Activation of Bypass Pathways

Acquired resistance to HER2 therapy has been associated with Src-mediated constitutive activation of the PI3K/AKT/mTOR and MAPK/ERK pathways. Src is a proto-oncogene involved in multiple signaling cascades that are implicated in tumor growth and metastasis [135]. Once activated, Src upregulates signals such as AKT, ERK, and STAT3 to increase cell proliferation, and interacts with adhesion proteins to accelerate cell migration and cell-cycle progression [136]. Additionally, there is evidence to suggest that Src may also be the link for crosstalk between EGFR and MET [114]. In preclinical studies, trastuzumab-resistant, lapatinib-resistant, and afatinib-resistant GC cells were found to have a significant increase in the basal levels of phosphorylated Src and ERK1/2 when compared to sensitive GC cells [114,115,128]. Treatment with bosutinib, a Src tyrosine kinase inhibitor, prevented downstream signaling and restored trastuzumab-sensitivity in previously resistant GC and biliary tract cancer cell lines [137].

Nuclear factor erythroid 2-related factor (NRF2) is a stress response mediator in solid tumors that functions by conferring inducible resistance to varying types of stress. When activated, NRF2 localizes to the nucleus to stimulate the expression of cytoprotective genes. Studies have shown that NRF2 can enhance progression and metastasis and lead to chemoradiotherapy resistance [138,139]. It is not fully understood how NF2 facilitates cancer growth, but it has been shown to augment the expression of genes in proliferative and metabolic pathways, such as PI3K/AKT/mTOR [140]. This could be related to RPS6, a substrate of the mTOR complex, that increases NF2 activity. NRF2 serves as a poor prognostic indicator, as patients with high levels have shorter PFS and worse response to trastuzumab in HER2-amplified GC patients. Similarly, in vitro studies of lapatinib- and trastuzumab-resistant GC cell lines were found to have elevated NRF2 protein expression [140].

### 4.4. Epithelial to Mesenchymal Transition (EMT)

EMT is another potential mechanism of acquired resistance to HER2-directed therapy in GI malignancies. Pre-clinical studies with trastuzumab-, lapatinib-, and afatinib-resistant GC cell lines revealed increased levels of mesenchymal markers, such as vimentin and testican-1, with a concurrent decrease in E-cadherin to promote an aggressive EMT phenotype [141,142,143]. These resistant cell lines were able to maintain MET activation and MAPK signaling despite treatment with lapatinib [143].

The HER4–YAP1–EMT axis has also been implicated in anti-HER2 therapy resistance. NRG1-activated HER4 can interact with YAP1 to induce EMT and trastuzumab resistance in GC cell lines [142,144,145]. Silencing HER4 inhibited EMT and PI3K signaling, leading to increased apoptosis of resistant cells and decreased tumor growth in vitro and in vivo [141].

### 4.5. HER2 Receptor Variants

HER2 splice variants lead to alternative receptors with impaired binding to HER2-directed agents. The 95kDa truncated HER2 fragment (p95HER2) is a well-studied variant in breast cancer that is also present in GC. p95HER2 does not have an extracellular domain, which results in trastuzumab and pertuzumab resistance. Lapatinib and other TKIs are still effective as they bind to the intracellular domain of HER2 [146].

The ERBB2d16 (HER2d16) isoform is a HER2 splice variant with a 16-amino-acid deletion in the juxta transmembrane domain. While better studied for its potential role in resistance in breast cancer, it has also been implicated in trastuzumab resistance in GC. Its presence is thought to promote tumor invasion, metastasis, and resistance because it retains the ability to form homodimers. HER2d16 homodimers activate multiple signaling cascades and changes that are involved in trastuzumab resistance, including NOTCH, Src, and EMT. An increased HER2d16/HER2 ratio in GC cell lines is associated with an EMT-like phenotype of higher vimentin and lower E-cadherin expression [147,148].

### 4.6. Masking of HER2 Epitopes

Mucin 4 (MUC4) is a membrane-bound mucin glycoprotein that is frequently expressed in gastric cancers and associated with HER2. The transmembrane subunit of MUC4 is believed to trigger HER2 phosphorylation through direct interaction with it [149,150]. Overexpression of MUC4 leads to HER2 activation and masking of the trastuzumab-binding site on HER2. It is not fully understood how MUC4 is upregulated in gastric cells, but evidence suggests a STAT3-centered positive feedback loop. Preclinical studies with GC cell lines have indicated that MUC4 expression increases in response to catecholamine stimulation via the STAT3 and ERK pathways [151]. Moreover, prolonged exposure to trastuzumab in GC cells potentiates STAT3 hyperactivation thus promoting the expression of MUC4. Consequently, increased MUC4 levels cause HER2 activation and decreased trastuzumab binding, ultimately resulting in acquired resistance to HER2-directed therapy [152].

### 4.7. ADC-Specific Resistance Mechanisms

Resistance to ADCs in GI malignancies remains poorly understood but is thought to be related to altered transport pathways and metabolism. In order to exert its antitumoral function, trastuzumab emtansine (T-DM1) binds to the HER2 receptor and is internalized by the target tumor cells [153]. This process is facilitated by the highly acidic environment created by vacuolar H+-ATPase (V-ATPase) for the hydrolases to function. Preclinical studies of GC cell lines demonstrated aberrant V-ATPase activity contributed to T-DM1 resistance by decreasing its metabolism to the active DM1. Interestingly, HER2-targeted ADCs with different linkers, such as a protease-cleavable linker, namely T-DXd, were able to overcome this form of resistance [154].

## 5. Future Perspectives

As resistance to conventional therapies continues to challenge the treatment of HER2-targeted GI cancers, the need for innovative approaches is critical. Sequential testing of various HER2 treatments to overcome resistance has been conducted. DESTINY-GASTRIC01 utilized T-DXd versus standard chemotherapy for HER2-positive advanced GC who progressed on prior therapies including trastuzumab, resulting in FDA approval [70]. Another trial studying the uses of ADCs includes the HERACLES RESCUE trial (NCT03418558), which is examining T-DM1 as monotherapy in patients with disease progression following lapatinib and trastuzumab treatment. These signal the ongoing efforts to find effective second-line options.

Further complicating the resistance landscape are MAPK pathway alterations, such as *MET* amplification or *PTEN* mutations. The NSABP FC-11 phase II study explores the dual targeting of the MAPK pathway using neratinib, a pan-HER inhibitor, with trastuzumab versus cetuximab in metastatic colorectal patients, showing promising early results [155]. Moreover, it has been previously reported that PD-L1 expression is regulated by both the MAPK and PI3K pathways as well as clinical studies illustrating enhanced antitumor immune response with T-DXd in breast cancer, suggests that patients with specific pathway alterations could benefit from combined PD-L1 and HER2 targeting [156,157,158,159]. This approach is currently being evaluated in trials such as SGNTUC-024 (NCT044307838) and KEYNOTE-811 (NCT03615326) as well as PANTHERA trial (NCT02901301), where combinations including pembrolizumab are showing potential PFS benefits in advanced GC [69,160]. Other mechanistically novel therapeutics are also underway, such as bispecific antibody targeting both PD-1 and HER2, as alternative approach in treatment paradigm [161].

HER2-specific vaccines are being explored as a strategy to address resistance to immunotherapy in HER2-amplified GC. The IMU-131/HER-Vaxx vaccine, containing HER2 B-cell epitopes fused with diphtheria toxoid and an adjuvant, has shown safety and immune responses in a phase I study in HER2-overexpressing gastric cancer [162]. Ongoing phase I and II trials are evaluating the efficacy of HER-Vaxx in patients diagnosed with metastatic or advanced HER2/neu-overexpressing GC/GEJ adenocarcinomas, either as a monotherapy or in combination with chemotherapy or immunotherapy [163].

Similar to the resistance seen for EGFR therapies, the HERACLES-A study revealed *PIK3CA* and *PTEN* alterations at the progression of HER2-targeted therapies, suggestive of acquired resistance mechanisms to HER2 targeted treatment [126,164,165]. In vitro studies using combinatory strategies, such as lapatinib with the PIK3 inhibitor copanlisib, which have shown synergistic effects against trastuzumab-resistant GC cell lines, underlining the potential and need for further clinical evaluation of combination strategies to overcome resistance to HER2 targeted therapies [166].

Lastly, advancements in theranostics, such as 89Zr-Trastuzumab PET/CT, are revolutionizing the non-invasive monitoring of HER2 expression and therapy response, facilitating the development of personalized treatment plans. Together with ctDNA analysis for real-time resistance detection, these technologies are steering HER2-positive GI cancer treatments towards more dynamically adaptable and personalized approaches.

Future research should focus on integrating HER2-targeted therapies with other modalities like chemotherapy, radiation, and immunotherapy to enhance treatment efficacy and manage resistance. Ongoing and future clinical trials exploring these synergistic combinations and novel therapies are critical to determining the most effective treatment regimens for improved outcomes for patients with HER2-positive gastrointestinal cancers. By continuing to unravel the complex role of HER2 in cancer, enhancing diagnostic accuracy, and developing more sophisticated treatment strategies, the potential for improving patient outcomes and quality of life remains substantial.

## 6. Conclusions

The exploration of HER2′s role in GI malignancies has significantly advanced our understanding of cancer progression and treatment, opening new therapeutic avenues for HER2-positive GI malignancies. The advent of HER2 targeted therapies such as trastuzumab, tucatinib, and T-DXd has transformed clinical outcomes, demonstrating the effectiveness of targeted treatment approaches. However, the variability in HER2 expression and classification as well as the challenge of various treatment resistance covered in this review highlight the complex nature of HER2 resistance in GI malignancies and necessitate ongoing advancements in diagnostic and therapeutic technologies.

Intratumoral heterogeneity and resistance to HER2-targeted therapies are major hurdles in the clinical management of HER2-positive cancers. The development of liquid biopsies and next-generation sequencing offers promising solutions, potentially allowing for real-time monitoring of HER2 mutations and tailored treatment adjustments. As these technologies evolve, they may enable more dynamic and responsive treatment plans, enhancing the efficacy of HER2-targeted treatments. Moreover, understanding the mechanisms of resistance—whether through genetic mutations, alternative signaling pathways, changes in the tumor microenvironment, or novel therapy specific escape pathways—is crucial for developing next-generation therapies that can overcome these challenges.

## Figures and Tables

**Figure 1 cancers-16-02854-f001:**
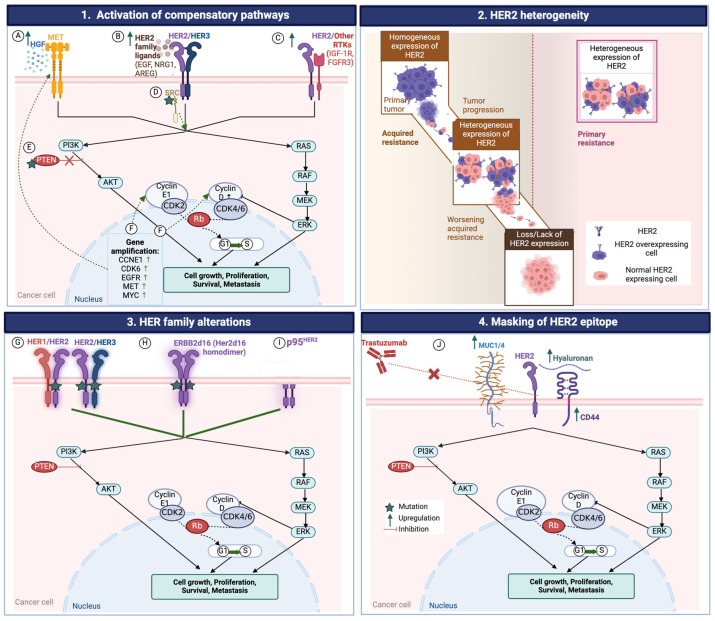
Select mechanisms of HER2 targeted resistance: (**1**) Acquired resistance to HER2 therapy involves compensatory activation of PI3K/AKT/mTOR and MAPK/ERK pathways, facilitated by various mechanisms such as MET amplification interacting with HGF [A], overexpression of HER2-family ligands (EGF, AREG, NRG1) [B], upregulation of phosphorylated IGF-1R [C], activation of Src-promoting signaling pathways [D], PTEN loss-of-function mutation causing PIP3 dysregulation [E], and co-amplifications of CDK6, CCND1, CCNE1 driving cell-cycle progression [F]. (**2**) The heterogeneous expression of HER2 in tumors contributes to primary and acquired resistance to HER2-targeted therapies. Progressive loss of HER2 positivity is a key factor in acquired resistance. Intratumoral heterogeneity is also common in gastric cancer, even before HER2 treatment (**3**) Mutations in HER2, together with HER2 and HER3 mutations, activate the PI3K–AKT pathway [G]. The generation of ERBB2d16 (HER2d16), a variant lacking exon 16′s extracellular domain, stabilizes homodimers and activates downstream signaling [H]. Overexpression of p95HER2 produces a truncated HER2 form without the extracellular domain [I] (**4**) Excessive mucin 4 (MUC4) expression and the CD44–polymeric hyaluronan complex can cause inhibition of Trastuzumab binding to HER2. Figure Abbreviations: AREG Amphiregulin; CCNE1 Cyclin E1 gene; CDK Cyclin; EGF/R Epidermal growth factor/receptor; FGFR3 Fibroblast growth factor receptor 3; HER1/2/3 Human epidermal growth factor rector 1/2/3; HGF Hepatocyte growth factor; IGF-1R Insulin-like growth factor 1 receptor; MET Mesenchymal epithelial transition; NRG1 Neuregulin 1; SRC Proto-oncogene tyrosine-protein kinase Src; PI3K/AKT phosphoinositide 3-kinase/AKT; PTEN phosphatase and tensin homolog; Ras/RAF/MEK/ERK Ras/Raf/MEK/extracellular signal-regulated kinase; Rb Retinoblastoma protein; RTK Receptor tyrosine kinase.

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
