# Peer review of "Resistance to Anti-HER2 Therapies in Gastrointestinal Malignancies"

_cancers, 2024, doi:10.3390/cancers16162854_

Round 1
Reviewer 1 Report
Comments and Suggestions for Authors
Mo et al has submitted a manuscript entitled “Resistance to Anti-HER2 Therapies in Gastrointestinal Malignancies” for consideration for publication. This review is a good overview of Anti-HER2 treatments, and these treatments are applied to GI cancers. Nevertheless, this review suffers from a lack of focus and differences from other recent reviews on this subject. List below are my specific comments.
1. This title suggests that the focus of the review will be on treatment resistance. Most of the review is on HER targeted treatments and how this is used in GI cancers.
2. In the resistance section, there was not enough context to GI cancers and too much on overall mechanisms of resistance.
3. There are many reviews on HER2 and GI cancers. How is this review different? If it is on the resistance, then it needs to be more focused.
4. A future perspective section needs to be added to give clarity as to where is the future of HER2 treatments in GI cancers needs to improve.
5. Too much of the review is focused on HER2 targeted treatments in general not specific to GI cancers. This has been reviewed extensively and does not added value to the review.
Comments on the Quality of English LanguageMinor errors in english.
Author Response
Dear Reviewer,
Thank you for your insightful comments. We have carefully addressed your concerns and appreciate your guidance.
- This title suggests that the focus of the review will be on treatment resistance. Most of the review is on HER targeted treatments and how this is used in GI cancers.
We have edited and shortened the section on HER2 targeted treatments while expanding on the mechanisms of resistance in GI malignancies.
2. In the resistance section, there was not enough context to GI cancers and too much on overall mechanisms of resistance.
The mechanisms we reviewed in the paper are related to GI malignancies based on preclinical studies. We have now included specific examples and references of how they were studied in gastric cancer and colorectal cancer cell lines to provide better context.
For your convenience, these are the lines: 403-406, 408-410, 425-428, 435-444, 450-454, 464-466, 468-472, 473-476, 486-488, 497-502, 510-513, 517-521, 523-525, 534-540, 547-552, and 558-562.
3. There are many reviews on HER2 and GI cancers. How is this review different? If it is on the resistance, then it needs to be more focused.
Prior reviews on HER2 resistance were focused primarily on breast cancer or general mechanisms of all HER2 resistance. Based on your previous comments, we have provided clear examples of HER2 resistance specifically for GI malignancies.
4. A future perspective section needs to be added to give clarity as to where is the future of HER2 treatments in GI cancers needs to improve.
We have included a future perspective section that discusses detecting mutations early, bispecific ADC, and combination therapy as potential methods to overcome resistance.
5. Too much of the review is focused on HER2 targeted treatments in general not specific to GI cancers. This has been reviewed extensively and does not added value to the review.
We edited to include examples of HER2 targeted treatments that are related to GI malignancies, such as those currently approved for use or undergoing clinical trials.
Reviewer 2 Report
Comments and Suggestions for Authors
The Authors beautifully describe the emerging anti-HER2 agents for different GI cancers, examine potential resistance pathways, and consider some aspects of HER2 expression interpretation and heterogeneity.
The review is well written.
The References are appropriate.
Lines 57-67: I think it would be helpful for the readers to have a deeper insight into the scoring system for HER2 expression, as the 1+ positivity is becoming a debatable problem (as in breast cancer). 1+ cases and 2=IHC without amplification by ISH are becoming a large coohort.
Line 259: please check for abbreviations and eventually explain what CPS is referred to (I think PD-L1 expression).
Paragraph 4.1 Heterogeneity. The differences between biopsy and surgical specimens in the interpretation of HER2 expression should be addressed: the heterogeneity is sometimes complex to assess on biopsy samples, and the cut-offs are different in determining positivity. Guidelines on biopsy sampling in gastric, GEJ, and colorectal cancers should be cited, as HER2 expression on biopsy material might be the only assessment done in metastatic settings..
Author Response
Dear Reviewer,
Thank you for your comments. We have carefully addressed your concerns as detailed below:
- Lines 57-67: I think it would be helpful for the readers to have a deeper insight into the scoring system for HER2 expression, as the 1+ positivity is becoming a debatable problem (as in breast cancer). 1+ cases and 2=IHC without amplification by ISH are becoming a large coohort.
We have included more information on the growing use of HER2-low in colorectal and gastric cancer in lines 70-74.
2. Line 259: please check for abbreviations and eventually explain what CPS is referred to (I think PD-L1 expression).
We have defined CPS (line 213).
3. Paragraph 4.1 Heterogeneity. The differences between biopsy and surgical specimens in the interpretation of HER2 expression should be addressed: the heterogeneity is sometimes complex to assess on biopsy samples, and the cut-offs are different in determining positivity. Guidelines on biopsy sampling in gastric, GEJ, and colorectal cancers should be cited, as HER2 expression on biopsy material might be the only assessment done in metastatic settings.
We have included a more thorough explanation of heterogeneity, guidelines for testing for both ESMO and ASCO, and guidelines for surgical/biopsy sampling (lines 345-358).
Reviewer 3 Report
Comments and Suggestions for Authors
Summary:
In the manuscript, the authors provide an extensive review of the challenges and advancements in combating resistance to HER2-targeted therapies in various gastrointestinal cancers. The manuscript is well-structured and provides a thorough overview of the subject.
Major concern:
Lack of Detailed Conclusion: The conclusion section is brief and lacks a detailed summary of the key points discussed in the manuscript. Expanding this section to include the main findings, their implications for clinical practice, and future research directions would strengthen the manuscript.
Minor concern:
Inconsistent Terminology: Ensure consistent use of terminology throughout the manuscript. For instance, the terms "trastuzumab-resistant", "trastuzumab-resistance", and "trastuzumab resistance" should be used consistently to avoid confusion.
Author Response
Dear Reviewer,
Thank you for comments and guidance. We have addressed your concerns as detailed below:
Major concern:
Lack of Detailed Conclusion: The conclusion section is brief and lacks a detailed summary of the key points discussed in the manuscript. Expanding this section to include the main findings, their implications for clinical practice, and future research directions would strengthen the manuscript.
We included a future perspectives section with potential ways to overcome resistance. We have also expanded on the conclusion.
Minor concern:
Inconsistent Terminology: Ensure consistent use of terminology throughout the manuscript. For instance, the terms "trastuzumab-resistant", "trastuzumab-resistance", and "trastuzumab resistance" should be used consistently to avoid confusion.
We have made edits to consistently use "trastuzumab-resistant" or "trasutuzmab resistance" depending on the correct grammar.
Round 2
Reviewer 1 Report
Comments and Suggestions for Authors
The revised manuscript is vastly improved. The future research paragraph in the conclusion should be in the future perspectives. Otherwise a complete review.
Comments on the Quality of English LanguageEnglish is fine.
Author Response
Dear Reviewer,
Thank you for your comments. We moved the future research paragraph to the future perspectives section.